# Compress, Gather, and Recompute: REFORMing Long-Context Processing in Transformers

Woomin Song[1,*],   Sai Muralidhar Jayanthi[2],   Srikanth Ronanki[2],
Kanthashree Mysore Sathyendra[2],   Jinwoo Shin[1],   Aram Galstyan[2],
Shubham Katiyar[2],   Sravan Babu Bodapati[2]

[1]KAIST,   [2]Amazon AGI

## Abstract

As large language models increasingly gain popularity in real-world applications, processing extremely long contexts, often exceeding the model's pre-trained context limits, has emerged as a critical challenge. While existing approaches to efficient long-context processing show promise, recurrent compression-based methods struggle with information preservation, whereas random access approaches require substantial memory resources. We introduce REFORM, a novel inference framework that efficiently handles long contexts through a two-phase approach. First, it incrementally processes input chunks while maintaining a compressed KV cache, constructs cross-layer context embeddings, and utilizes early exit strategy for improved efficiency. Second, it identifies and gathers essential tokens via similarity matching and selectively recomputes the KV cache. Compared to baselines, REFORM achieves over 52% and 34% performance gains on RULER and BABILong respectively at 1M context length. It also outperforms baselines on $\infty$-Bench, RepoEval, and MM-NIAH, demonstrating flexibility across diverse tasks and domains. Additionally, REFORM reduces inference time by 30% and peak memory usage by 5%, achieving both efficiency and superior performance.

## 1 Introduction

The ability to handle extremely long contexts, often exceeding the original model's pre-trained context limits, has emerged as a critical challenge for the advanced usage of large language models (LLMs) in real-world scenarios. This capability is essential for various applications, such as processing life-long user interactions, understanding and debugging repository-level codebases, and handling multi-modal inputs (interleaved sequences of text and visual information can result in extremely long contexts). However, under existing Transformer-based language model architectures [1, 2], processing such long sequences often causes significant computational challenges, requiring substantial computation as well as memory resources. These demanding requirements often prove infeasible in practical deployment settings, necessitating new technologies that can handle extremely long sequences with reasonable computational resources.

Current approaches to efficient context window extrapolation broadly fall into two categories: recurrent context processing and random access mechanisms. Recurrent methods [3, 4, 5, 6] divide the input into manageable chunks and iteratively process them while maintaining a summarized representation of prior chunks, typically by compressing or evicting parts of the Key-Value (KV) cache. While these approaches reduce memory and computational costs, they often suffer from 'forgetting' due to the loss of critical information during compression and/or eviction.

---

*Work done during an internship at Amazon.

39th Conference on Neural Information Processing Systems (NeurIPS 2025).

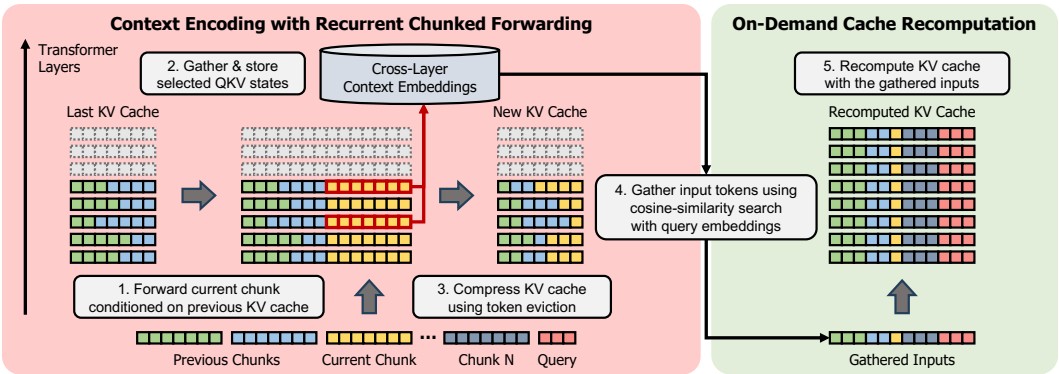

Figure 1: **An overview of the proposed framework.** REFORM efficiently processes long inputs through two phases. In the recurrent chunked forwarding phase, it segments inputs into chunks and processes them iteratively. In each iteration, REFORM (1) forwards each chunk conditioned on the previous KV cache, (2) extracts key QKV states from selected layers and heads for constructing cross-layer context embeddings, and (3) compresses the cache via token eviction [4]. An early exit strategy skips upper layers beyond those used for embedding collection, further improving efficiency. In the on-demand cache recomputation phase, REFORM selects important tokens via similarity search with the query embeddings (last part of the input), gathers them, and recomputes the KV cache for further generation.

In contrast, another line of work aims to enable dynamic random-access to the previous inputs by preserving the full KV cache and retrieving relevant portions when processing new chunks [7, 8]. These methods provide more flexibility in accessing prior context, as they allow selective re-attention to specific segments of the input. However, maintaining the full KV cache requires substantial memory resources, often leading to significant memory overhead and latency increases, especially in practical deployments where CPU memory offloading is necessary. Furthermore, the increased flexibility does not necessarily lead to high retrieval performance. These limitations highlight the need for a more balanced approach that combines efficiency with precise long-context handling.

To address the above challenges, we propose REFORM (**RE**current chunked **F**orwarding with **O**n-demand cache **R**eco**M**putation), a novel inference framework that combines the efficiency of recurrent approaches with the superior recall capabilities of random-access methods via a computationally efficient compress-gather-recompute pipeline. In contrast to existing recurrent methods that use compressed KV cache for generation, REFORM uses *compression* to construct and store lightweight token-wise embeddings of the input. Given a query, REFORM then uses these embeddings to *gather* most relevant input tokens via similarity matching, and *recomputes* the full KV cache for those tokens. This process yields high-fidelity yet efficient representations for query-relevant tokens, leading to a superior retrieval ability of the method while still benefiting from reduced memory overhead.

Figure 1 illustrates the overall approach. In the encoding phase, we process input tokens in chunks through an adaptive caching mechanism called recurrent chunked forwarding: as each chunk is processed, tokens are added to the KV cache and compressed by retaining only the heavy hitters (most influential tokens). Using this progressively sparsified KV cache, we compute representations up to an intermediate transformer layer L, collecting QKV states from multiple layers and heads to generate and store lightweight cross-layer context embeddings for all tokens. This multi-faceted efficiency strategy—combining chunked processing, sparse KV cache updates, and early exit—significantly reduces both computation time and memory overhead, as we maintain only small representations for retrieval while dynamically managing KV cache sparsity.

In the recomputation phase, the query tokens (corresponding to the recent context) identify relevant historical tokens through similarity matching with the stored retrieval embeddings, and only these selected tokens undergo full KV cache recomputation across all layers. While this phase requires full computation for selected tokens, this recomputation is crucial: it restores high-fidelity representations for contextually important tokens, ensuring accurate processing of long-range dependencies. By selectively recomputing only the most relevant tokens, we achieve a better balance between computa-

tional efficiency and model performance, allowing detailed historical context access while avoiding the costs of maintaining full representations for all tokens.

Our extensive evaluations demonstrate REFORM's effectiveness across various long-context understanding tasks. In needle-in-a-haystack tests, REFORM achieves perfect recall for contexts up to 1 million tokens at various depths. On more complex benchmarks, REFORM significantly outperforms existing methods, achieving over 52% performance gain on RULER and 34% on BABILong at 1M context lengths with the Mistral-NeMo-Instruct-2407 [2] model, compared to the best-performing baselines. On ∞-Bench, REFORM achieves 50.2% average accuracy with the same model, substantially exceeding the baseline performance of 37.6%. REFORM also outperforms the baselines in RepoEval, scoring 65.3% performance with Qwen2.5-Coder-1.5B-Instruct on API-level code completion while the best baseline gives 61.8%.

Operating at the transformer architecture level, REFORM is modality-agnostic and applicable to any domain/modality the base model supports. To demonstrate its flexibility, we evaluate REFORM on three multi-modal needle-in-a-haystack datasets, achieving 57.51% performance with the Pixtral-12B-2409 [9] model, exceeding the baseline performance of 52.95%.

Finally, REFORM delivers substantial efficiency improvements over recent state-of-the-art long-context processing methods. Compared to InfLLM [7] and InfiniPot [6], REFORM reduces inference time by 80% and 33% and peak memory usage by 32% and 5% respectively, in evaluations with 256k token inputs. These results demonstrate that REFORM effectively combines the benefits of both recurrent compression and random access approaches while mitigating their respective limitations.

## 2   Related Works

In this section, we discuss the existing approaches for extending LLM's native context window to efficiently handle extremely long inputs. We categorize these approaches into two groups: methods that use recurrent context processing and methods that leverage random access. We provide a more comprehensive discussion on related works in Section B.

**Recurrent context processing.** To address the computational challenges of long-context processing, several studies explore the use of recurrence for greater efficiency. A line of works [10, 11, 12] introduce architectural changes to Transformers, enabling chunk-level recurrence operations to process long contexts in smaller, manageable units. However, these approaches typically necessitate extensive training of the model, and therefore is not directly applicable to existing pre-trained large language models. More recent efforts leverage KV cache eviction to iteratively encode input chunks and compress the KV cache, avoiding architectural modifications or altering the model parameters. For instance, StreamingLLM [3] maintains fluent generation by preserving initial and most recent tokens while compressing intermediate ones. Later approaches [4, 5, 6] identify important tokens from the prior context, enabling more informative cache compression. Despite their efficiency, the process of compressing prior inputs often results in the loss of critical information, leading to 'forgetting' issues. Consequently, these methods may struggle with tasks requiring precise retrieval of earlier inputs. REFORM addresses this issue through its gather and recompute phases, which yields high-fidelity representation of all query-relevant input tokens.

**Random access approaches.** An alternative direction is to enable random access to prior context, akin to full attention, but in a more computationally efficient manner. These methods typically store the full KV cache in memory and dynamically retrieve relevant tokens as needed. Some approaches train the model [13] or an auxiliary side-network [14] to utilize the retrieved tokens effectively. More recently, several strategies achieve the same goal without modifying the model parameters, by storing the full KV cache in memory and retrieving it dynamically [7, 8]. While these methods allow random access to any part of the input sequence, they introduce significant memory overhead due to the need to maintain large caches. In practice, this often necessitates CPU offloading, which can further increase latency. Furthermore, the flexibility to access previous context may not necessarily lead to high retrieval performance. In contrast, REFORM uses KV cache compression and constructs compact token-level embeddings using only the high-performing heads to reduce memory overhead while still maintaining high retrieval performance.

# 3 Method

In this section, we present the details of our proposed method. In Section 3.1, we first describe REFORM's recurrent chunk forwarding phase in detail. This phase efficiently constructs token-level, cross-layer context embeddings by segmenting the long input into multiple chunks and repeatedly processing them while conditioning on a compressed previous KV cache. In Section 3.2, we further elaborate on how we construct the cross-layer context embeddings. Finally, in Section 3.3, we describe how we use the context embeddings to identify the relevant input segments and highlight our on-demand cache recomputation framework that enables random access to previous contexts while maintaining the integrity of the KV cache. We outline the full procedure in Figure 1 and Algorithm 1.

## 3.1 Embedding Extraction with Recurrent Chunked Forwarding and Early-Exit

Encoding long contexts with pre-trained Transformers is often infeasible due to the quadratic computational cost and the model's limited context window. To overcome this problem, we focus on recurrent KV cache compression approaches that allow the processing of infinite context under limited resources and context windows. Here, we describe the encoding process in detail, discuss key efficiency benefits, and present our early exit strategy that provides further efficiency gains when using recurrent chunked forwarding to create context embeddings.

**Embedding extraction with recurrent chunked forwarding.** To process extremely long inputs under limited computational budget and context windows, we adopt an iterative KV cache compression approach [3, 4, 5, 6]. Specifically, we segment the long input into larger chunks (32k tokens for our experiments) and apply KV cache compression after forwarding each chunk to better utilize parallel computation. For KV cache compression, we employ attention-based token eviction following H2O [4]. After compression, we reassign the position IDs so that the tokens in the compressed cache have consecutive position IDs. This position reassignment allows the model to handle longer sequences beyond its pre-trained context limit.

Unlike existing approaches that aim to directly use these compressed KV cache for generation, we use it only to construct context embeddings that will later be used to identify which part of the input is required for generating a response.

**Early exit.** Utilizing a compressive approach for creating embeddings introduces an additional benefit: efficiency can be further improved by employing an early exit strategy. As observed in Section 3.2, high-performing embeddings are often available in the lower Transformer layers. Therefore, forwarding the inputs through the remaining layers after the topmost layer used for embedding extraction is unnecessary. The proposed early exit strategy reduces both computation and memory requirements because we do not need to keep the KV cache for the upper layers.

## 3.2 Constructing Cross-Layer Context Embeddings

Prior research has revealed the existence of specialized Transformer heads distributed across different layers that can accurately retrieve relevant information from long context input [15]. To construct informative embeddings, we thus analyze the retrieval performance of various heads and embeddings in Transformers to determine the most suitable ones for our method. Specifically, we compare the token-level retrieval performance of attention scores (without positional encoding, to make it applicable to extremely long inputs), and cosine similarity between hidden states, or the attention QKV states (i.e. embeddings resulting from QKV projection in attention layers) across Transformer layers.

Table 1: **Comparing similarity search methods.** Best-3 MNR scores (lower is better) corresponding to different similarity search methods, including attention, and cosine similarity search using hidden states (HS) or attention QKV states. Scores are measured with Mistral-Nemo-Instruct-2407, and averaged over 500 multi-hop QA examples.

| Type | Dim. | Top-1 | Top-2 | Top-3 | Avg. |
|---|---|---|---|---|---|
| Attention | 160 | 6.91 | 7.70 | 7.81 | 7.47 |
| Cosine-HS | 5120 | 9.40 | 9.63 | 9.80 | 9.61 |
| Cosine-Q | 160 | 6.48 | 6.74 | 6.93 | 6.72 |
| Cosine-K | 160 | 6.77 | 7.31 | 7.41 | 7.16 |
| Cosine-V | 160 | 5.77 | 6.57 | 6.57 | 6.30 |

**Embedding head identification.** We conducted a set of experiments on a multi-hop question-answering dataset to identify the best embeddings to use for the similarity search; see Section C.1 for details. We report the MNR scores for the top-performing layers and heads in Table 1. Somewhat

surprisingly, we observe that cosine similarity search with attention QKV states often outperform the widely-used attention scores. It also outperformed using cosine similarity between the hidden states despite having a much smaller size. This finding suggests that, with careful selection of the appropriate heads, directly using the QKV states from the attention layer and using them for cosine similarity search can achieve a very high performance, while requiring minimal memory.

Figure 2 illustrates the distribution of the top-performing value heads. Notably, the best-performing heads are not necessarily from the final Transformer layers. This observation implies that forwarding inputs through the upper layers may not be required for constructing effective retrieval embeddings, serving as a key motivation for our early exit strategy described in the previous section.

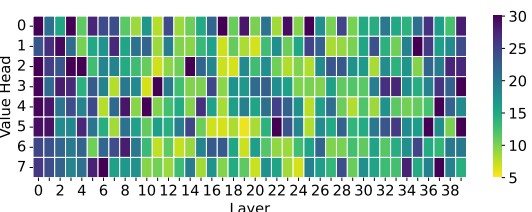

Figure 2: **MNR Scores for Value Heads.** The distribution of the MNR scores (lower is better) across value states of different attention heads, measured by Mistral-Nemo-Instruct-2407 model for 500 synthetic multi-hop QA examples. 256-token heavy hitter budget was used for computation.

To create a universal embedding that is useful for a wide range of tasks, we do additional experiments to identify heads that can effectively represent complex input patterns. Specifically, we create a synthetic key-value retrieval task, which involves embedding multiple sentences of the format *"The value corresponding to the id {key} is {value}."* within the WikiText [16] corpus, where keys and values are random 10-character ASCII strings.

We selected the top-performing embeddings for each synthetic dataset after evaluating 500 samples each. We highlight that although head selection is based on relatively short synthetic data (8k tokens), the benefits extrapolate to longer contexts involving millions of tokens.

**Combining multiple heads.** After identifying the top-performing heads, we combine their embeddings to create a single, token-level embedding. In our preliminary experiments, we observed that using an average of retrieval scores obtained from different heads often improves final retrieval performance. Accordingly, we concatenate the gathered embeddings after normalizing them:

$$e_{\texttt{comb}} = \texttt{concat}\left(\left\{\frac{e_{\texttt{i}}}{||e_{\texttt{i}}||}, \texttt{i} \in \texttt{selected\_heads}\right\}\right)$$

This approach ensures that performing a cosine similarity search using the resulting embedding is mathematically equivalent to independently computing cosine similarity scores for each head and then averaging them.

### 3.3 On-Demand Cache Recomputation

To enable random access to the previous inputs, we utilize the cross-layer context embeddings to identify the input segments that are relevant to the last part of the input. Then, we gather the corresponding input embeddings and forward them through the model again, re-constructing the KV cache with the most relevant inputs. We conduct the detailed process as follows.

**Identification of significant inputs.** After constructing the cross-layer context embeddings corresponding to the input, we perform a token-level similarity search between the query (the last part of the input) and the remaining inputs using the context embeddings. Then, we max-pool the similarity scores over the query to tokens to ensure that each token is assigned a single score. To preserve the continuity of the identified inputs, we further max-pool each token's score with the 128 adjacent tokens. After processing the significance scores, we identify the tokens with the highest scores. We always keep the initial and final 256 tokens to maintain coherence.

**On-Demand Cache Recomputation.** Once we identify the relevant segments, we gather the corresponding input embeddings and forward them through the model again, recomputing the KV cache. The new KV cache is then used for the further decoding process. By introducing an on-demand cache recomputation scheme, we avoid the need of storing the full KV cache while enabling random access to previous inputs, significantly reducing the memory requirements.

# 4  Experiments

This section demonstrates the performance of our method across diverse tasks. In Section 4.1, we begin by showcasing the precise retrieval ability of our approach using a needle-in-a-haystack benchmark. In Section 4.2, we evaluate our method on RULER [17] and BABILong [18], which are more advanced synthetic datasets consisting of more complex long-context assesments, including more challenging needle-in-a-haystack tasks, aggregation tasks, question answering, and multi-hop reasoning. In Section 4.3, we further test REFORM's performance on more realistic tasks including ∞-Bench [19] and RepoEval [20], as well as highlighting the flexibility of our approach by evaluating it on multi-modal benchmarks. In Section 4.4, we compare our approach to retrieval-augmented generation, an emerging direction for handling long inputs. Finally, we ablate on the key components and analyze the efficiency of our approach in Section 4.5.

**Common setup and baselines.** Throughout the paper, we mainly compare our approach against context extrapolation methods that do not alter the model parameters, with a primary focus on recurrence-based and random-access approaches. Specifically, we compare our method against StreamingLLM [3], TOVA [5], H2O [4], InfiniPot [6], and InfLLM [7]. We also include a truncation baseline, which simply drops the middle part of the input. For H2O, we restrict attention score computations to the last 128 tokens of each chunk for efficient implementation.

For all text-based experiments, we use Mistral-NeMo-Instruct-2407 [2] and Qwen2.5-7B-Instruct [21] models. For code completion experiments, we use Qwen2.5-Coder-1.5B and 7B models. For multi-modal experiments, we use Pixtral-12B-2409 [9]. All recurrent baselines operate with a KV cache budget and chunk size of 32k tokens, and InfLLM also uses 32k active KV cache budget. We always keep the initial and recent 256 tokens in cache for all baselines and REFORM to maintain the coherency of the text. For REFORM, we use a recomputation budget of 8k tokens for Mistral-Nemo-Instruct-2407 and 16k tokens for all other models. We provide more details in Section C.1.

## 4.1  Needle-In-A-Haystack Evaluation

To evaluate the precise retrieval performance of our approach, we employ the Needle-in-a-Haystack (NIAH) benchmark [22]. In this task, a specific "needle" sentence (*"The best thing to do in San Francisco is eat a sandwich and sit in Dolores Park on a sunny day"*) is embedded within various depths of irrelevant context consisting of diverse essays by Paul Graham. The model must correctly answer the question: *"What is the best thing to do in San Francisco?"* For evaluation, we consider a response to be correct if it contains all three key phrases: *"eat a sandwich"*, *"sit in Dolores Park"*, and *"a sunny day."*



Figure 3: **Needle-In-A-Haystack Evaluation.** We visualize the retrieval accuracy of Qwen2.5-7B-Instruct at different depth and context lengths. Performance is averaged over 20 samples.

In Figure 3, we measure the performance of our method at different context lengths and needle depths. Our method demonstrates perfect performance across all setups up to 1M tokens, highlighting our method's robustness in handling extremely long contexts while maintaining precise retrieval performance.

## 4.2  Performance on RULER and BABILong

In this section, we further demonstrate the performance of our approach in more diverse and challenging synthetic benchmarks. Specifically, we evaluate different methods on an extended version of the RULER [17] and BABILong [18] benchmarks. RULER is a synthetic long-context benchmark consisting of diverse and challenging needle-in-a-haystack tasks, as well as some aggregation and question answering tasks. BABILong further challenges the model by introducing more difficult tasks, such as multi-hop reasoning. Although the original version of RULER only supports up to 128k tokens, we further extend the dataset to 1M using the same recipe to evaluate on longer inputs.

Table 2: **Evaluation on RULER and BABILong.** We measure the performance on an extended version of the RULER [17] and BABILong [18] benchmark. We report the averaged performance of all tasks at different context lengths. The best values are highlighted in **bold**.

| | RULER | | | | | | | BABILong | | | | |
|---|---|---|---|---|---|---|---|---|---|---|---|---|
| | 64k | 128k | 200k | 300k | 400k | 500k | 1M | 64k | 128k | 256k | 512k | 1M |
| *Mistral-Nemo-Instruct-2407* | | | | | | | | | | | | |
| Truncation | 32.6 | 20.4 | 17.8 | 15.2 | 12.3 | 12.5 | 10.8 | 32.2 | 26.2 | 17.0 | 13.6 | 14.0 |
| StreamingLLM | 27.6 | 13.8 | 11.6 | 9.3 | 7.2 | 7.1 | 4.7 | 38.8 | 23.4 | 15.4 | 11.0 | 6.2 |
| TOVA | 21.6 | 15.3 | 14.0 | 11.8 | 7.9 | 8.7 | 4.6 | 37.8 | 23.6 | 14.4 | 9.6 | 3.4 |
| H2O | 15.1 | 7.4 | 7.8 | 5.6 | 4.2 | 5.7 | 3.6 | 38.0 | 25.2 | 16.2 | 7.2 | 3.6 |
| InfiniPot | 26.9 | 19.4 | 15.6 | 14.5 | 12.7 | 13.4 | 12.0 | 39.6 | 26.8 | 18.6 | 11.2 | 8.8 |
| InfLLM | 52.7 | 39.7 | 28.5 | 24.9 | 20.9 | 22.0 | 23.3 | 40.6 | 34.0 | 23.6 | 13.0 | 9.6 |
| **REFORM (Ours)** | **79.9** | **81.1** | **83.0** | **84.6** | **84.1** | **83.5** | **75.5** | **57.4** | **51.4** | **50.6** | **47.6** | **48.8** |
| *Qwen2.5-7B-Instruct* | | | | | | | | | | | | |
| Truncation | 46.3 | 25.1 | 21.8 | 17.4 | 14.9 | 15.2 | 11.3 | 48.4 | 33.4 | 27.4 | 20.0 | 15.6 |
| StreamingLLM | 43.5 | 25.3 | 18.7 | 17.3 | 11.8 | 11.8 | 9.1 | 53.4 | 40.6 | 33.2 | 23.8 | 19.6 |
| TOVA | 66.2 | 27.7 | 25.7 | 25.8 | 21.9 | 20.4 | 17.0 | 56.0 | 46.6 | 40.6 | 29.4 | 21.8 |
| H2O | 51.8 | 20.9 | 18.5 | 17.1 | 11.6 | 12.1 | 8.7 | 57.0 | 41.6 | 36.4 | 24.6 | 18.8 |
| InfiniPot | 65.7 | 51.7 | 39.2 | 33.9 | 27.8 | 26.7 | 23.7 | 59.6 | 51.0 | 53.4 | 48.2 | 40.2 |
| InfLLM | 47.1 | 34.2 | 29.2 | 24.0 | 22.0 | 23.2 | 23.8 | 43.0 | 29.2 | 20.4 | 15.4 | 11.4 |
| REFORM (Ours) | **78.2** | **75.8** | **74.7** | **74.9** | **74.9** | **73.0** | **75.1** | **61.6** | **60.4** | **59.8** | **58.8** | **58.8** |

We highlight the evaluation results on RULER and BABILong in Table 2. In both benchmarks, REFORM outperforms the baselines by a large margin, indicating its superiority in tasks that require precise recall of essential parts of the context, benefiting both from the ability to locate essential contexts from long inputs and the removal of distribution shifts in the KV cache that commonly come with recurrence-based or random-access approaches.

## 4.3 Performance on ∞-bench, RepoEval, and Multi-Modal Evaluations

In this section, we further evaluate the performance on more diverse long context handling tasks. Specifically, we evaluate the performance of REFORM on ∞-bench [19], a more realistic long-context benchmark including tasks derived from long books and dialogues, and RepoEval [20], a repository-level code completion benchmark, to demonstrate that REFORM is useful in realistic tasks. Furthermore, we highlight the broad applicability of REFORM by demonstrating its performance on a multi-modal benchmark, MM-NIAH [23].

Table 3: **Evaluation on ∞-Bench.** We evaluate each method on more 10 datasets from ∞-Bench [19]. We did not evaluate on C.Run and M.Calc datasets since no method was capable of achieving a nonzero score with these models. The best values are highlighted in **bold**.

| | R.PK | R.Num | R.KV | En.Sum | En.QA | En.MC | En.Dia | Zh.QA | C.Debug | M.Find | Avg. |
|---|---|---|---|---|---|---|---|---|---|---|---|
| *Mistral-Nemo-Instruct-2407* | | | | | | | | | | | |
| Truncation | 27.1 | 21.4 | 3.6 | 13.7 | 16.0 | 51.1 | 11.5 | 25.2 | 28.9 | 20.3 | 21.9 |
| StreamingLLM | 28.1 | 15.3 | 0.0 | 12.5 | 12.6 | 45.9 | 6.5 | 19.2 | 27.2 | 0.0 | 16.7 |
| TOVA | 82.2 | 47.0 | 0.0 | 12.3 | 13.8 | 47.2 | 8.0 | 6.6 | 25.1 | 4.6 | 24.7 |
| H2O | 31.5 | 9.5 | 0.0 | 14.2 | 17.6 | 49.3 | 6.0 | 21.2 | 26.1 | 15.7 | 19.1 |
| InfiniPot | 84.1 | 13.6 | 0.0 | 11.9 | 17.1 | 52.0 | 7.0 | 11.3 | 27.4 | 15.7 | 24.0 |
| InfLLM | 100.0 | 100.0 | 1.0 | 16.9 | 17.4 | 58.1 | 7.0 | 24.5 | 24.1 | 27.1 | 37.6 |
| REFORM (Ours) | 100.0 | 100.0 | 88.2 | 18.2 | 18.0 | 70.3 | 18.5 | 26.7 | 25.9 | 36.0 | **50.2** |
| *Qwen2.5-7B-Instruct* | | | | | | | | | | | |
| Truncation | 27.1 | 27.1 | 7.4 | 29.0 | 13.3 | 43.2 | 15.0 | 9.34 | 37.1 | 45.7 | 25.4 |
| StreamingLLM | 28.8 | 28.8 | 6.0 | 29.2 | 8.6 | 52.4 | 14.5 | 9.51 | 32.5 | 28.6 | 23.9 |
| TOVA | 100.0 | 100.0 | 1.2 | 29.4 | 8.6 | 56.8 | 15.0 | 10.65 | 34.3 | 42.6 | 39.8 |
| H2O | 93.1 | 85.4 | 0.0 | 31.0 | 11.0 | 56.3 | 15.5 | 11.97 | 34.8 | 44.6 | 38.4 |
| InfiniPot | 100.0 | 99.8 | 0.8 | 30.6 | 11.3 | 59.0 | 17.0 | 9.99 | 36.6 | 44.9 | 41.0 |
| InfLLM | 100.0 | 99.8 | 1.6 | 27.6 | 9.6 | 38.0 | 12.0 | 10.41 | 29.7 | 45.1 | 37.4 |
| REFORM (Ours) | 100.0 | 100.0 | 32.8 | 27.8 | 16.5 | 61.6 | 21.5 | 11.81 | 33.0 | 21.7 | **42.7** |

Table 4: **Evaluation on RepoEval and MM-NIAH.** For RepoEval, we report the edit similarity (ES) score on RepoEval api-level completion task and line-level completion task with 1.5B and 7B models. For MM-NIAH, we report normalized performance across input lengths to ensure equal contribution from each context length range. We do not run multi-modal evaluation for InfLLM, as its implementation only supports text-based models. Best results are in **bold**.

| Method | RepoEval | | | | MM-NIAH | | | |
|---|---|---|---|---|---|---|---|---|
| | 1.5B API | 1.5B Line | 7B API | 7B Line | Retrieval | Counting | Reasoning | Avg. |
| Truncate | 54.8 | 63.9 | 59.2 | 59.5 | 72.2 | 18.7 | 51.2 | 47.4 |
| StreamingLLM | 55.0 | 62.7 | 59.9 | 58.4 | 71.9 | 17.8 | 49.8 | 46.5 |
| TOVA | 54.7 | 62.2 | 59.7 | 59.8 | 82.9 | 18.8 | 54.1 | 52.0 |
| H2O | 55.1 | 63.4 | 61.2 | 59.6 | 83.3 | 18.9 | 53.5 | 51.9 |
| InfiniPot | 59.4 | 68.4 | 66.2 | 63.8 | 85.4 | 18.8 | 54.7 | 53.0 |
| InfLLM | 61.8 | 66.8 | 64.3 | 66.3 | N/A | N/A | N/A | N/A |
| **REFORM (Ours)** | **65.3** | **72.4** | **68.7** | **69.4** | **89.2** | **22.0** | **61.3** | **57.5** |

For ∞-bench, we evaluate both Mistral-Nemo-Instruct-2407 and Qwen2.5-7B-Instruct models. For RepoEval, we perform evaluation using code-specific models, namely Qwen2.5-Coder-1.5B/7B-Instruct. For each sample, we provide the entire repository as the context except for the file that is being completed for the given sample. We report the edit similarity (ES) score as the evaluation metric. Finally for multi-modal evaluations, we use Pixtral-12B-2409 [9].

As shown in Table 3 and Table 4, REFORM consistently outperforms all baselines in all three benchmarks. This highlights REFORM's superior performance on realistic tasks, and its flexibility to handle diverse inputs, even across different modalities.

## 4.4    Comparison to Retrieval Augmented Generation

We now compare REFORM to Retrieval Augmented Generation (RAG), a popular method for processing long inputs [24, 25]. RAG frameworks segment inputs into smaller chunks, which are independently encoded, and use external retrieval models to identify relevant segments. While effective in some scenarios, RAG suffers from key limitations.

First, REFORM avoids the context fragmentation inherent in RAG by conditioning retrieval embeddings on the entire input, ensuring global context continuity and allowing for cohesive processing of long contexts. Second, while RAG frameworks are constrained by the training domain of the retrieval model—requiring domain-

Table 5: **Comparison with RAG.** We compare the performance of RAG methods and REFORM on four groups of needle-in-a-haystack datasets (single, multikey, multivalue, and multiquery) from RULER at 300k contexts, using Mistral-NeMo-Instruct-2407 model.

| | Single | M.Key | M.Value | M.Query |
|---|---|---|---|---|
| Sparse RAG | 86.7 | 77.3 | 88.5 | 90.0 |
| Dense RAG | 87.3 | 57.3 | 82.5 | 78.0 |
| REFORM | **99.3** | 93.3 | 98.5 | **100.0** |
| + RAG | **99.3** | **94.7** | **99.0** | **100.0** |

specific retraining or advanced adaptations for different domains and modalities—REFORM is inherently flexible and can seamlessly handle diverse domains, including multi-modal applications, without requiring such modifications. Finally, REFORM integrates retrieval functionality directly into the model, eliminating the need for external retrieval models.

In Table 5, we compare the performance of REFORM against RAG approaches using sparse and dense retrievers on the needle-in-a-haystack datasets from RULER at 300k contexts. We provide a more detailed experiment setup in Section C.4. REFORM consistently outperforms both approaches in all evaluations, demonstrating its robustness and efficiency. Furthermore, we explore a hybrid approach by combining REFORM with a dense retriever, blending REFORM 's token-level significance scores with retrieval scores using a weighted sum (25% for the dense retriever, 75% for REFORM). This approach performs even better, highlighting the complementary strengths of REFORM and RAG.

Table 6: **Ablation study and efficiency analysis.** (a) We report the average performance on RULER 300k and BABILong 512k datasets using Mistral-NeMo-Instruct-2407 model. (b) We compare the inference time and peak memory usage required for generating 10 tokens conditioned on 256k inputs. All measurements are made with the Mistral-NeMo-Instruct-2407 model on a single H100 GPU, and are averaged over 10 samples. The best values are highlighted in **bold**.

<table>
<tr><td colspan="3" align="center">(a) Ablation study.</td></tr>
<tr><td></td><td>RULER</td><td>BABILong</td></tr>
<tr><td>REFORM (Ours)</td><td>**84.6**</td><td>**47.6**</td></tr>
<tr><td>w/ StreamingLLM</td><td>82.7</td><td>44.6</td></tr>
<tr><td>w/ TOVA</td><td>81.4</td><td>46.8</td></tr>
<tr><td>w/ Random heads</td><td>80.3</td><td>43.0</td></tr>
<tr><td>w/ Worst heads</td><td>44.7</td><td>22.8</td></tr>
<tr><td>w/ Kernel size 5</td><td>18.4</td><td>36.8</td></tr>
<tr><td>w/ Kernel size 17</td><td>39.4</td><td>45.8</td></tr>
</table>

<table>
<tr><td colspan="3" align="center">(b) Efficiency Analysis.</td></tr>
<tr><td></td><td>Time (s)</td><td>Memory (GB)</td></tr>
<tr><td>StreamingLLM</td><td>36.58</td><td>37.34</td></tr>
<tr><td>H2O</td><td>41.33</td><td>37.85</td></tr>
<tr><td>TOVA</td><td>39.46</td><td>37.06</td></tr>
<tr><td>InfiniPot</td><td>40.90</td><td>37.06</td></tr>
<tr><td>InfLLM</td><td>129.14</td><td>51.62</td></tr>
<tr><td>**REFORM (Ours)**</td><td>**27.24**</td><td>**35.00**</td></tr>
</table>

## 4.5 Ablation Studies

We conduct an ablation study to evaluate the key components contributing to the effectiveness of our approach. Specifically, we analyze the impact of (1) the choice of the recurrent compression method, (2) the selection of attention heads used for retrieval, and (3) size of the maxpool kernel applied during the gather stage. We assess model performance on the BABILong and RULER benchmarks at context lengths of 512k and 300k, respectively.

**Choice of recurrent compression method.** To demonstrate the generality of our approach, we replace our recurrent compression component with alternative methods, namely StreamingLLM and TOVA. While H2O yields the best results, Table 6a shows that other compression methods achieve comparable performance. This further highlights the flexibility of our framework and its potential for even higher performance with more advanced compression techniques.

**Choice of attention heads.** To examine the importance of attention head selection for embedding construction, we replace the selected heads with (1) four randomly chosen heads and (2) four worst heads, identified based on poor performance on both synthetic datasets used for head selection. As shown in Table 6a, the heads selected by our mechanism achieve the best performance, demonstrating its effectiveness. Random heads generally show lower but reasonable performance. In contrast, using bad heads results in a substantial performance drop on both benchmarks, underscoring the importance of proper attention head selection to ensure effective embedding construction.

**Pooling kernel size.** In the on-demand cache recomputation phase, we apply max-pooling over 129-token windows to smooth token-level similarity scores. As demonstrated in Table 6a, reducing the pooling size to 5 or 17 tokens significantly degrades performance, highlighting the importance of pooling to maintain contextual information during the recomputation.

## 4.6 Efficiency Analysis

To highlight the efficiency benefits of our approach, we measure the peak memory usage and inference time required for processing a long input. We outline the results in Table 6b. InfLLM suffers from high inference time due to frequent memory transfer between CPU and GPU and requires large memory to store the cache. Recurrent methods offer faster inference at lower memory costs, enjoying the benefits of using a fixed-size KV cache. Our approach shows even lower latency and memory requirements compared to the recurrent baselines thanks to the early exit, which saves computation as well as memory by removing the need to keep the KV cache for the upper layers. We present further latency breakdown analysis in Section D.2, highlighting that REFORM keeps both time to first token (TTFT) and time per output token (TPOT) minimal.

# 5 Conclusion

We introduce REFORM, a novel inference framework for efficient long-context processing. REFORM incrementally processes input chunks while maintaining a compressed KV cache, extracting key QKV states to construct cross-layer context embeddings. An early-exit strategy enhances efficiency, and a similarity-based selection mechanism identifies and gathers essential tokens for KV cache recomputation. REFORM outperforms existing methods across long-context benchmarks while reducing inference time and memory usage. Furthermore, its modality-agnostic design makes it applicable to a wide range of use cases including multi-modal applications.

**Limitations and Future work.** One limitation is that the effectiveness of REFORM depends in part on the gather stage's ability to identify informative tokens. While our results across diverse benchmarks demonstrate that this step is generally reliable, edge cases may exist where the model's selection could be further optimized. Another limitation is the redundant attention score computation in the current implementation. Our current implementation recomputes attention scores using matrix multiplication to compute the eviction scores, as Flash Attention does not output attention weights. Integrating eviction score computation into the Flash Attention kernel could further improve REFORM's efficiency.

Also, we directly adopt the token eviction criteria proposed by H2O [4] as the compression component of our framework. As a future direction, we plan to investigate more sophisticated compression approaches tailored for constructing the context embeddings. Furthermore, applications to more diverse data modalities such as audio and video is another promising direction to explore.

## Acknowledgements

For WS and JS: This work was supported by Institute for Information & communications Technology Promotion(IITP) grant funded by the Korea government(MSIT) (No.RS-2019-II190075 Artificial Intelligence Graduate School Program (KAIST); No. RS-2024-00509279, Global AI Frontier Lab; RS-2022-II220959, Few-shot Learning of Causal Inference in Vision and Language for Decision Making).

WS expresses his sincere gratitude to Jihoon Tack for his insightful discussions, support, and mentorship throughout the course of this work.

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

# A    REFORM algorithm

We illustrate the overall process of REFORM through a pseudocode in Algorithm 1.

---

**Algorithm 1** Overview of REFORM

---

    **procedure** FORWARDCHUNK(chunk, cache, emb)
        /* Initialize hidden states */
        $hs \leftarrow$ input
        /* Forward with early exit */
        **for** layer **in** model_layers[:early_exit_layer] **do**
            $hs, cache, qkv \leftarrow$ layer.Forward($hs, cache$)
            /* Save selected embeddings */
            emb.SaveSelected(qkv)
        **end for**
        /* Evict less important tokens */
        $cache \leftarrow$ Compress($cache$)
        **return** cache, emb
    **end procedure**
    **procedure** REFORM(input)
        /* Initialize */
        $cache, emb \leftarrow$ EmptyInit()
        /* Prepare input chunks */
        $context, query \leftarrow$ SplitQuery(input)
        $chunks \leftarrow$ ChunkInputs(context) + [query]
        /* Recurrent chunked forwarding */
        **for** $c_i$ **in** chunks **do**
            $cache, emb \leftarrow$ ForwardChunk($c_i, cache, emb$)
        **end for**
        /* Gather relevant inputs */
        $relevant\_inputs \leftarrow$ GatherRelevant(input, emb)
        /* On-demand recomputation */
        $cache \leftarrow$ model.Forward(relevant_inputs)
        **return** cache
    **end procedure**

---

# B    Extended related works

**Extending LLMs to handle extremely long inputs.** To extend the context windows of Large Language Modles (LLMs) efficiently, various approaches have been proposed. A significant body of work focuses on modifying positional embeddings. These include scaling Rotary Positional Embeddings (RoPE) [26] beyond the model's context limit [27, 28, 29, 30], applying attention masks [31], or adjusting the relative distances between tokens to fall within a predefined range [32, 33]. Another line of research explores fine-tuning techniques to adapt models for longer contexts [34, 35]. While these methods enable models to handle extended inputs, they do not address the significant computational and memory costs introduced by the self-attention mechanism, limiting their practical utility for extremely long contexts. Hence, we did not include them as baselines in our experiments.

**Other approaches for efficient long context processing.** Together with the recurrent KV cache compression approaches, a large volume of recent works focus on reducing the size of the KV cache to enable more efficient inference at long contexts. For example, SnapKV [36] proposes to forward the full input through the model, and then compress the cache by evicting tokens based on attention scores. While efficient at decoding-time, it requires the model to first process the full input, and therefore is not applicable to extremely long inputs that exceed the model's pre-trained context window. Alternatively, HOMER [37] proposes to use a hierarchical divide-and-conquer approach to combine the encoding and eviction process. Some works propose to further enhance KV cache compression by merging tokens instead of evicting them [38, 39], but their experiments also only consider inputs within the model's context limit, and their extrapolation capabilities remain unknown. Some recent works propose another direction to keep the full cache only for some selected attention

heads known as 'retrieval heads' [40, 41], reducing the memory burden of preserving the full KV cache. Other works investigate quantization [42, 43] and low-rank cache compression [44] to further reduce the memory requirements of the KV cache. However, these methods also cannot extrapolate to longer sequences beyond the model's pre-trained context limit.

## C    Experimental Details

### C.1    Evaluating Retrieval Heads and Embeddings

**Dataset preparation.** To evaluate the embeddings, we constructed a synthetic dataset based on multi-hop question answering. In this setup, we embedded documents from the HotPotQA dataset [45] at random positions within a long text corpus derived from the WikiText dataset [16]. Each question was appended at the end of the context, and token-level labels were created, where tokens from the golden documents were marked as ground truth. All samples were designed to be 8k tokens long, which is within the context window of the Mistral-7B-Instruct-v0.2 model.

**Embedding extraction.** To simulate long-context scenarios where full attention computation is infeasible due to computational or memory constraints, we employed a recurrent chunk forwarding method based on H2O [4], elaborated in Section 3.1. For attention, we compute the retrieval scores using the dot product between query states (Q) and the key states (K) without applying positional encoding. For all other embeddings, we compute the significance scores using cosine similarity between question embeddings and context embeddings, followed by max-pooling over question tokens. Additionally, retrieval scores for each context token were smoothed by mean-pooling with 20 neighboring tokens.

**Performance measurement.** Retrieval performance was quantified using the Mean Normalized Rank (MNR), which is calculated as the average normalized rank of the golden tokens. Lower scores correspond to higher performance, as the golden tokens have a high rank.

$$\texttt{MNR} = \frac{1}{\texttt{len(gold\_doc)}} \sum_{t \in \texttt{gold\_doc}} \frac{\texttt{rank(t)}}{\texttt{num\_tokens}}$$

### C.2    Detailed Head Selection Process

**Step 1: Data preparation.** We construct two synthetic tasks for attention head selection: a pattern matching task and a multi-hop question answering task. For the pattern matching task, we insert sentences into the WikiText corpus with the format: "The value corresponding to the id key is value." where both the key and value are random 10-character ASCII strings. For the multi-hop QA task, we use passages from HotPotQA that contain the information required to answer a given question. The tokens corresponding to this key information are labeled as "golden tokens."

**Step 2: Context encoding.** We apply recurrent chunked forwarding using H2O to each sample in the dataset, extracting token-level QKV embeddings from every layer and attention head.

**Step 3: Significance score computation.** For each QKV head in each layer, we compute a token-wise significance score. This is done by (1) calculating cosine similarity between context tokens and question tokens, (2) aggregating the similarity scores over the question tokens via max-pooling, and (3) smoothing the scores using mean pooling across a 20-token window.

**Step 4: Performance measurement and head selection.** We evaluate the effectiveness of each attention head using the Mean Normalized Rank (MNR) score, as described in Section C.1. Based on these scores, we select four heads: two that perform best on the pattern matching task, and two that perform best on the multi-hop QA task.

### C.3    Multimodal Evaluations

**Baseline details.** In the multi-modal experiments, we evaluate the model performance using recurrence-based methods only, as the codebase for InfLLM only supports text-based models. For InfiniPot, the NuC (novelty under compression) score cannot be utilized for cache compression for multi-modal models because the vision tokens do not output a logit. Therefore, we only apply the CaP (catalyst prompt) score for the InfiniPot baseline in multi-modal experiments.

## C.4 Comparison against RAG

**Experiment setup.** We follow the setup in OP-RAG [25] for the RAG experiments, segmenting the inputs to 128-token chunks and preserving the order of the chunks instead of rearranging them according to the retrieval scores. We use Mistral-NeMo-Instruct-2407 as the base LLM. We use BM25 [46] as the sparse retriever, and bge-large-en-v1.5 [47] as the dense retriever. For each sample, 8k tokens are retrieved in total, matching the KV size with our approach to ensure fair comparison.

## C.5 Efficiency Measurements

**Experiment setup.** We measure the average inference time and peak memory usage for generating 10 tokens conditioned on 256k tokens. All measurements are made on a single H100 GPU, and we apply Flash Attention 2 [48] for all measurements. We further elaborate the experiment setup for InfLLM, as the inference speed and memory consumption can largely vary depending on the configuration. We use the default configuration provided in their GitHub repository, while modifying the number of retrieved blocks to keep 32k active tokens in the cache. The maximum number of blocks cached in GPU was set to be the twice as large as the number of retrieved blocks, following the convention in their official configuration file.

## C.6 Embedding Construction and Similarity Search for REFORM

**Embedding head selection.** We construct the context embeddings by combining four QKV embeddings, where two heads are identified using the pattern matching dataset and the other two are identified using the multi-hop QA dataset. To balance between performance and efficiency gains, we select the top-performing heads from layers with depth under 70% for pattern matching heads. See Section D.1 for a more detailed discussion.

For Mistral-NeMo-Instruct-2407, the following heads are used:

1. Query head 9 at layer 15
2. Value head 5 at layer 19
3. Value head 0 at layer 27
4. Value head 7 at layer 27

For Qwen2.5-7B-Instruct, the following heads are used:

1. Value head 3 at layer 7
2. Key head 0 at layer 14
3. Value head 3 at layer 14
4. Value head 0 at layer 19

For Qwen2.5-Coder-1.5B-Instruct, the following heads are used:

1. Query head 3 at layer 8
2. Value head 1 at layer 11
3. Key head 0 at layer 14
4. Value head 0 at layer 15

For Qwen2.5-Coder-7B-Instruct, the following heads are used:

1. Value head 2 at layer 13
2. Key head 0 at layer 14
3. Value head 3 at layer 14
4. Query head 4 at layer 14

For Pixtral-12B-2409, the following heads are used:

1. Value head 3 at layer 10

2. Value head 5 at layer 19

3. Value head 0 at layer 27

4. Value head 7 at layer 27

**Similarity search.** REFORM performs a cosine similarity search between each token in the query (the final part of the input) and the remaining tokens. For better precision in identifying the relevant inputs, we remove the special tokens and the generation prefix (e.g. 'the answer is') when computing the similarity scores.

## D    Additional Results

### D.1    Embedding Head Identification for Pattern Matching Task

Table 7: **Comparing different LLM embeddings.** Best-3 MNR scores (lower is better) corresponding to the hidden states and the attention states, measured by Mistral-Nemo-Instruct-2407. Scores are averaged over 500 synthetic pattern matching examples.

| Type | Dim. | Top-1 | Top-2 | Top-3 | Avg. |
|---|---|---|---|---|---|
| Hidden States | 5120 | 1.72 | 1.88 | 2.10 | 1.90 |
| Attention | 160 | 1.24 | 1.36 | 1.37 | 1.32 |
| Query | 160 | 1.51 | 1.56 | 1.57 | 1.55 |
| Key | 160 | 1.53 | 1.65 | 1.72 | 1.63 |
| Value | 160 | 0.93 | 0.95 | 1.13 | 1.00 |

In this section, we present the distribution of MNR scores measured with our pattern matching dataset, similarly to what we presented in Table 1 and Figure 2. The corresponding results for pattern matching dataset is presented in Table 7 and Figure 4. The retrieval performance of QKV heads often outperform that of the hidden states, similarly to the case of multi-hop QA datasets.

Interestingly, the distribution of best-performing heads show a different pattern compared to the milti-hop QA dataset, and the heads at lower layers and middle-to-upper layers show the highest performance. This suggests that different heads show different characteristics depending on the task. It also motivates our approach of using the embeddings identified by the different tasks as it yields more general representations and makes similarity-based retrieval more accurate. It is also important to note that while the upper layer has more good-performing heads, these heads can be also identified in the mid-lower layers (e.g., Layer 16, Head 1). To balance the performance with the efficiency gains provided by early-exit strategy, we select the best-performing pattern-matching heads from layers under 70% of depth. This strategy ensures that we utilize the high-performing heads as well as enjoying the computation savings from early exit.

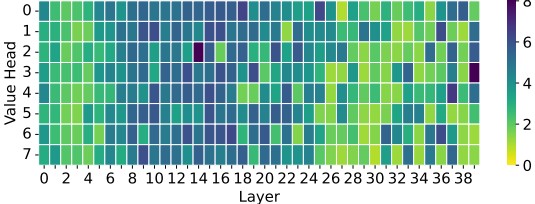

Figure 4: **MNR Scores for Value Heads.** The distribution of the MNR scores (lower is better) across value states of different attention heads, measured by Mistral-Nemo-Instruct-2407 model over 500 synthetic pattern matching examples. Recurrent chunked forwarding with 256-token heavy hitter budget was employed for computing the embeddings.

## D.2  Latency Breakdown

Here, we extend our latency analysis to three context lengths (only 256k in the main text), and separately present the pre-fill and decoding latency in Table 8a and Table 8b.

In Table 8a, InfiniPot and InfLLM shows slower processing speed compared to StreamingLLM due to the additional computation associated with dynamic cache eviction and retrieval. On the other hand, REFORM consistently shows faster prefill time compared to all baselines, thanks to the early-exit optimization. Further latency breakdown of the Compress+Gather stage and Recompute stage suggests that the recomputation overhead is minimal, taking under 1.52% of the total prefill time.

Table 8b shows a similar trend for the decoding time. InfiniPot and InfLLM shows slower decoding time compared to StreamingLLM due to the additional operations. On the other hand, REFORM also shows improved decoding speed as the generation is conditioned on KV cache created with a small recomputation budget (8k), unlike the baselines that condition on the full compressed cache (32k). (Note that these budgets are the hyperparameters used for the Mistral-Nemo experiments in our main text.)

Table 8: **Latency Breakdown.** (a) Time to first token (seconds) measurements and (b) time per output token (seconds) measurements (Mistral-Nemo-Instruct-2407, single H200, averaged over 20 runs and 200 tokens generated per measurement).

<table>
<tr><td colspan="4">(a) Time to first token (seconds).</td><td colspan="4">(b) Time per output token (seconds)</td></tr>
<tr><td>Model</td><td>256k</td><td>512k</td><td>1M</td><td>Model</td><td>256k</td><td>512k</td><td>1M</td></tr>
<tr><td>StreamingLLM</td><td>30.59</td><td>68.22</td><td>143.57</td><td>StreamingLLM</td><td>0.111</td><td>0.111</td><td>0.111</td></tr>
<tr><td>InfiniPot</td><td>35.77</td><td>73.67</td><td>149.64</td><td>InfiniPot</td><td>0.256</td><td>0.256</td><td>0.256</td></tr>
<tr><td>InfLLM</td><td>95.71</td><td>213.23</td><td>474.96</td><td>InfLLM</td><td>0.259</td><td>0.267</td><td>0.329</td></tr>
<tr><td>**REFORM (Ours)**</td><td>**26.24**</td><td>**53.68**</td><td>**108.64**</td><td>**REFORM (Ours)**</td><td>**0.040**</td><td>**0.040**</td><td>**0.040**</td></tr>
<tr><td> - Compress + Gather</td><td>25.84</td><td>53.28</td><td>108.24</td><td></td><td></td><td></td><td></td></tr>
<tr><td> - Recompute</td><td>0.40</td><td>0.40</td><td>0.40</td><td></td><td></td><td></td><td></td></tr>
</table>

## D.3  Evaluation with a Larger Model

In this section, we present additional results using the Qwen2.5-32B-Instruct model on the RULER and BABILong benchmarks. As shown in Table 9, REFORM consistently outperforms both H2O and InfiniPot across all evaluation settings, demonstrating the scalability and robustness of our method on larger models.

Table 9: **Performance of Qwen2.5-32B-Instruct.** We report the performance on key long-context benchmarks for H2O, InfiniPot, and REFORM. The best values are highlighted in **bold**.

| | RULER
Single 300k | RULER
Multikey 300k | RULER
Multivalue 300k | RULER
Multiquery 300k | BABILong
256k |
|---|---|---|---|---|---|
| H2O | 38.7 | 2.7 | 14.0 | 6.0 | 31.0 |
| InfiniPot | 69.3 | 19.3 | 40.0 | 74.0 | 54.2 |
| **REFORM (Ours)** | **100.0** | **90.0** | **96.0** | **100.0** | **67.6** |

# E  Further Discussions

## E.1  Complexity Analysis

**Time and memory complexity.** Recurrent baselines such as StreamingLLM have a time complexity of $O(L)$, where $L$ is the input length, assuming a fixed (or bounded) chunk size and query length. Each recurrence step involves a constant amount of computation, and the total number of steps scales linearly with the input length. Their memory complexity is $O(1)$, since only a fixed-size KV cache is maintained throughout.

In contrast, random-access baselines like InfLLM incur a time complexity of $O(L^2)$, primarily due to periodic cache lookups across the full input length. These methods also require $O(L)$ memory, as they store the entire KV cache in memory. This memory burden is significant, often requiring CPU offloading, which further increases latency.

REFORM maintains a time complexity of $O(L)$ through its recurrent chunked processing. Although it has an $O(L)$ memory cost due to storing token-level embeddings, these embeddings are very small in practice. Moreover, the early-exit strategy significantly reduces memory and compute requirements. Consequently, REFORM achieves faster speed and lower memory usage than both baselines (as demonstrated in Table 6b), highlighting the practical efficiency of our method.

**Amount of offline computation.** While REFORM involves offline computation for head selection, the computational overhead of head selection is minimal in practice. The selection is performed only once using synthetic inputs of 8k tokens, and the chosen heads are reused across all inference runs, independent of the downstream task or input domain. By selecting heads using short inputs and reusing them across long-context inference, REFORM keeps the additional computation minimal. Quantitatively, the entire head selection procedure requires roughly the same amount of computation (in FLOPs) as processing just two 1M-token inputs with H2O.

### E.2  Theoretical Analysis on Head Selection

Our attention head selection is motivated by findings from mechanistic interpretability literature, which show that different heads specialize in different functions [49, 50]. Some heads are more useful than others for tasks like pattern recognition or reasoning. We leverage this diversity by selecting only a few high-performing heads, both for better identification performance and to keep the size of token-level embeddings tractable (including all heads would be memory-intensive). Our method thus aims for functional specialization as well as computational efficiency.

Although we use cosine similarity rather than attention weights to evaluate heads, our approach shares key insights with existing interpretability work. Prior studies show that lower-layer heads often detect syntactic or structural features, mid-layer heads handle semantic reasoning, and upper-layer heads guide output generation [51]. This aligns with our empirical findings: the most useful heads for pattern matching tend to appear in lower and upper layers, while those for multi-hop QA (i.e. semantic understanding) concentrate in the mid-layers. These trends are reflected in our MNR visualizations (Figure 2 and Figure 4). Also, Skean et. al. [52] suggests that the middle layer representations are more useful than the final layer representations for 32 MTEB (Massive Text Embedding Benchmark, [53]) tasks, similar to what we observe for the multi-hop QA task.

Furthermore, our choice of the head identification tasks is rooted in information retrieval literature, which shows that hybrid retrievers [54, 55] combining sparse (structural) and dense (semantic) retrievers outperforms either alone. Inspired by this, we use two distinct head evaluation tasks: pattern matching and multi-hop QA. The former identifies heads that capture structural similarity, while the latter targets heads that encode semantic relevance. This dual-task strategy helps us select a complementary set of heads that are robust across diverse long-context scenarios.

## F  Broader Impacts

We believe that the high capability and flexibility will aid everyday use of large foundation models, by extending the model capabilities to efficiently and effectively handle very long contexts. On the other hand, such capabilities of REFORM could potentially enable malicious parties to analyze vast amount of data, enhancing the capabilities of autonomous systems that could be used for manipulation or misinformation.

## G  License Information for Datasets and Models

Here, we provide the license for all datasets and models used in our experiments. Apache 2.0 license is applied for Babilong, RULER, Mistral-Nemo-Instruct-2407, Qwen2.5-Coder family, and Pixtral-12B-2409. BSD license is also applied for some parts of Babilong dataset. MIT license is applied to Needle-in-a-Haystack, InfiniteBench, RepoEval, WikiText, and bge-large-en-v1.5. CC BY-SA 4.0 is applied for HotPotQA.

