# OpenReview forum: "Compress, Gather, and Recompute: REFORMing Long-Context Processing in Transformers"
_NeurIPS.cc/2025/Conference — NeurIPS 2025 poster_

### Official Review · Reviewer_cHLb · 2025-06-08

**Clarity:** 3
**Significance:** 2
**Originality:** 2
**Rating:** 3
**Confidence:** 3

**Summary:**

The paper studies architectural designs (without training) to improve model's memorization capability. Building on existing works, the paper proposes REFORM, which performs KV cache compression, extracting QKV states and recompute essential tokens on demand. The authors test their method on various benchmarks where the method shows significant improvement compared to not only existing methods but also competitive technology such as RAG. The ablation studies ablate on the state choice, compression choice as well as kernel size to select most similar tokens.

**Questions:**

Compared to H2O, can the authors clarify what major contribution the current work brings please?

**Ethical Concerns:**

["NO or VERY MINOR ethics concerns only"]

**Final Justification:**

I appreciate authors' rebuttal for providing clear answers to my concerns. However, I would like to have my scores unchanged.

As the authors explained in the rebuttal, compared to compress method like H2O, having the "gather-recompute" part significantly improves the end performance. However, 500xCompressor compared to ICAE clearly shows the advantage of using KV cache and there exists also quite abundant token selection literature that the paper builds on for the "gather-recompute" stage.

Due to the above reasons, the paper's claim would be only totally grounded when cited and ablated through each one of the contribution which I judge the paper is slightly lacking.

**Limitations:**

The paper mentioned transparently that directly using H2O as a limitation (although it is more like a performance bottleneck). I think the paper would benefit from a more deep and insightful limitation discussions.

**Quality:**

2

**Strengths And Weaknesses:**

Strength: The paper cites relevant works from both recurrent context processing and random access approaches which situate well the current's work contribution. The paper evaluates on many benchmarks with two open source LLMs which makes the contribution convincing.

Weakness: Compare to state of the art compression techniques H2O, the paper seems to mainly contribute on better state choosing and pooling which makes the paper have relatively weak technical contributions.

---

> ### Author Rebuttal · Authors · 2025-07-30
>
> Dear reviewer cHLB,
>
> We sincerely appreciate your efforts in reviewing our manuscript. We respond to each comment in the following content.
>
> ---
> **[W1, Q1] Comparison to H2O**
>
> As also noted by Reviewer NqoB, the key novelty of our work lies in proposing the **compress-gather-recompute pipeline**, which enables **effective and efficient, training-free context extension for large foundation models**. Below, we clarify the main distinctions from H2O:
>
> ---
> **Key Difference 1: REFORM overcomes the context forgetting problem inherent in H2O**
>
> Recurrent approaches like H2O compress input segments incrementally, which often leads to **context forgetting**, particularly when handling longer inputs. This issue arises because information from earlier segments may be lost during successive compressions, harming performance on tasks that require long-range context.
>
> To address this, REFORM introduces a compress-gather-recompute pipeline. The 'compress' stage encodes long input into token-level embeddings. Then, the 'gather' stage identifies important tokens for generating a response. Finally, the 'recompute' stage reconstructs the KV cache based on selected tokens. This design allows the model to **access relevant information regardless of its position in the input**, thus overcoming the forgetting issue.
>
> Thanks to this fundamental difference in the method's capability, **REFORM significantly outperforms H2O across diverse models and benchmarks**.
>
> ***Table 1.** Performance comparison with H2O (Llama-3.1-8B-Instruct).*
>
> |  | RULER 300k | RULER 1M | BABILong 256k | BABILong 1M | Infinite-Bench |
> |---|:---:|:---:|:---:|:---:|:---:|
> | H2O | 25.9 | 13.8 | 28.0 | 16.6 | 39.7 |
> | **REFORM (Ours)** | **82.0** | **79.0** | **46.0** | **43.8** | **53.2** |
>
> ***Table 2.** Performance comparison with H2O (Mistral-Nemo-Instruct-2407).*
>
> |  | RULER 300k | RULER 1M | BABILong 256k | BABILong 1M | Infinite-Bench |
> |---|:---:|:---:|:---:|:---:|:---:|
> | H2O | 5.6 | 3.6 | 16.2 | 3.6 | 19.1 |
> | **REFORM (Ours)** | **84.6** | **75.5** | **50.6** | **48.8** | **50.2** |
>
> ***Table 3.** Performance comparison with H2O (Qwen2.5-Coder-1.5B/7B-Instruct).*
>
> |  | RepoEval API (1.5B) | RepoEval Line (1.5B) | RepoEval API (7B) | RepoEval Line (7B) |
> |---|:---:|:---:|:---:|:---:|
> | H2O | 55.1 | 63.4 | 61.2 | 59.6 |
> | **REFORM (Ours)** | **65.3** | **72.4** | **68.7** | **69.4** |
>
> ***Table 4.** Performance comparison with H2O on MM-NIAH (Pixtral-12B-2409).*
>
> |  | Retrieval | Counting | Reasoning | Average |
> |---|:---:|:---:|:---:|:---:|
> | H2O | 83.3 | 18.9 | 53.5 | 51.9 |
> | **REFORM (Ours)** | **89.2** | **22.0** | **61.3** | **57.5** |
>
> ---
> **Key Difference 2: Different usage of recurrent KV cache compression**
>
> While H2O uses recurrent compression directly to generate new tokens from the compressed KV cache, **REFORM leverages the compression differently**. We use it to generate **token-level embeddings**, which are later employed to select relevant tokens for recomputation. The suggested approach is **orthogonal to the specific compression mechanism** (e.g., H2O, TOVA, StreamingLLM), and empirically performs well with compression mechanisms other than H2O, as shown in Table 6(a) of the main text.
>
> Such a different usage **enables early-exit optimization**, as embeddings only require QKV states from selected layers (see L144–L149). H2O, by contrast, requires KV caches from all layers to be used for generation. Therefore, the early-exit optimization cannot be used for H2O.
>
> Thanks to the early-exit optimizations, **REFORM is more efficient than H2O**, both in terms of computation and memory. Specifically, our efficiency analysis in Table 6(b) from the main text suggests that **REFORM reduces the inference time by 34.1% and peak memory usage by 7.53% compared to H2O**.
>
> ---
>
> We believe these technical differences and empirical improvements clearly demonstrate that REFORM offers substantial advancements over H2O. Additionally, the proposed pipeline outperforms newer baselines like InfiniPot and InfLLM in both accuracy and efficiency, as demonstrated throughout the main text.
>
> ---
> **[L1] Deeper discussion on limitations.**
>
> Thanks for the constructive suggestions regarding more deep and insightful limitation discussions. Here, we provide some additional limitations of our work.
>
> One limitation is that the effectiveness of REFORM depends in part on the gather stage’s ability to identify informative tokens. While our results across diverse benchmarks demonstrate that this step is generally reliable, edge cases may exist where the model's selection could be further optimized.
>
> Another limitation is the redundant attention score computation in the current implementation. Our current implementation recomputes attention scores using matrix multiplication to compute the eviction scores, as Flash Attention does not output attention weights. Integrating eviction score computation into the Flash Attention kernel could further improve REFORM’s efficiency.

---

### Official Review · Reviewer_gpyK · 2025-07-03

**Clarity:** 3
**Significance:** 2
**Originality:** 2
**Rating:** 4
**Confidence:** 3

**Summary:**

This paper introduces REFORM, an inference framework to handle long contexts by combining ideas from two lines of long-context processing works -- those that process chunks of context to create "summary" KV caches, and those that compress the KV cache by only retaining relevant tokens.
Experiments are on RULER and Babilong datasets, as well as infinity bench and repoEval.

**Questions:**

1. The presentation of how attention heads are chosen is slightly confusing.

2. What is the performance of vanilla attention? It is unclear what the drop in performance is to full attention. While I appreciate that this is probably not possible to compute for very long contexts, it would be useful to report this for the 64k context lengths.

 3. Comparison to missing baselines would be helpful to calibrate the strength of this method.

**Ethical Concerns:**

["NO or VERY MINOR ethics concerns only"]

**Final Justification:**

The author rebuttal addressed my questions about the complexity analysis (which i hope can be made clearer in the manuscript) and comparison to baselines.

I will reiterate that, according to me, this paper is makes incremental progress in an already very crowded field of KV cache compression and long context modeling.

**Limitations:**

Yes.

**Quality:**

2

**Strengths And Weaknesses:**

Strengths:
1. Interesting approach combining ideas from two parallel lines of work.
2. The proposed approach shown improvement over the baselines reported in the paper.

Weaknesses:
1. Missing Baselines: the paper omits comparisons with strong baselines, such as SnapKV, DuoAttention, Minference, MoA, CEPE etc. Without comparisons with these baselines, it is hard to evaluate the strength of this approach.

2. The paper does not include any discussion of complexity for the proposed approach and baselines. Furthermore, no implementation details are included. Is the method compatible with flash attention? REFORM uses H2O for token eviction, which I believe is incompatible. How does this affect latency. (related question: are the wall clock times in table 6b reported w/ or w/o flash attention?)

---

> ### Author Rebuttal · Authors · 2025-07-30
>
> Dear reviewer gpyK,
>
> We sincerely appreciate your efforts in reviewing our manuscript. We respond to each comment in the following content.
>
> ---
> **[W1] Clarification on Research Objective**
>
> We would like to emphasize that our main research goal is to propose an efficient, **training-free context window extension method** for large foundation models (see L18–L19, L28, L87, L226–L231). We will revise the manuscript to more clearly state this objective.
>
> Hence, the suggested baselines, i.e., sparse attention variants such as DuoAttention, MInference, and MoA **address a different problem**, as they aim to improve efficiency **within the model's native context window** (e.g., up to 128k tokens for LLaMA-3.1-8B-Instruct). Due to the reason, we did not compare them in the paper as they cannot be used for inputs that are longer than the model's context limit. Nevertheless, they are orthogonal to our approach and could potentially be combined with REFORM to further enhance efficiency.
>
> While CEPE is capable of extending the context window, **it requires training the bidirectional encoder and cross-attention layers**, deviating from our training-free objective. Furthermore, REFORM will likely outperform CEPE as it is one of the early efforts on context extension, first released in February 2024. Notably, Figure 4 of the CEPE paper shows that it struggles on simple needle-in-a-haystack evaluations, whereas REFORM maintains 100% accuracy even at 1M tokens.
>
> Finally, we remark that SnapKV, while originally designed for inference within the native context window, can be adapted to longer inputs by applying it iteratively over input chunks (as suggested by Reviewer NqoB). As shown in Table 1, **REFORM consistently outperforms such a variant of SnapKV**.
>
> We believe that these discussions further clarify REFORM’s positioning in the long-context inference space, and will add them to the revised manuscript.
>
> ***Table 1.** Performance comparison with recurrent SnapKV variant (Llama-3.1-8B-Instruct).*
>
> |  | RULER 300k | RULER 500k | RULER 1M | BABILong 256k | BABILong 512k | BABILong 1M | Infinite-Bench |
> |---|:---:|:---:|:---:|:---:|:---:|:---:|:---:|
> | SnapKV | 62.5 | 59.1 | 57.8 | 36.0 | 31.6 | 25.6 | 43.7 |
> | **REFORM (Ours)** | **82.0** | **82.0** | **79.0** | **46.0** | **46.6** | **43.8** | **53.2** |
>
>
>
> ---
> **[W1, Q2,  Q3] Comparison with Baselines Within the Context Limit**
>
> Although our focus is on extending beyond the context window, we agree that evaluating REFORM within the context limit provides valuable insight, especially given that there are several open-source models that natively support 1M tokens. Table 2 compares REFORM with full attention, DuoAttention, and Minference using the Llama-3-8B-Instruct-Gradient-1048k model (1M-token support).
>
> Somewhat surprisingly, **REFORM consistently outperforms sparse baselines and often surpasses full attention**, especially at longer context lengths. This is likely due to the gather-and-recompute stage, which filters out irrelevant information, making the task easier for the model to solve.
>
> In Table 3, we further compared the performance of REFORM and full attention on Llama-3.1-8B-Instruct and Mistral-Nemo-Instruct-2407. We observed similar trends, where REFORM matches or outperforms full attention, and the gap becomes larger at longer contexts. While our main goal is in context extension beyond the model's context limits, these results suggest that **REFORM can be useful even within the native context window** by selectively handling only the relevant inforamtion.
>
> ***Table 2.** Performance within the context limit (Llama-3-8B-Instruct-Gradient-1048k). Default configurations from the official repositories were used for Duo Attention and MInference. RULER-Multi corresponds the average accuracy of the multikey, multivalue, and multiquery tasks.*
>
> ||RULER Single 128k|RULER Single 300k|RULER Multi 128k|RULER Multi 300k|Babilong 128k|Babilong 256k|
> |-|:-:|:-:|:-:|:-:|:-:|:-:|
> |Full Attention|**100.0**|**100.0**|84.4|73.3|34.2|28.0|
> |Duo-Attention|**100.0**|**100.0**|83.2|73.4|33.6|25.8|
> |MInference|**100.0**|**100.0**|84.4|72.9|33.8|26.4|
> |**REFORM (Ours)**|**100.0**|**100.0**|**98.0**|**95.7**|**37.8**|**37.4**|
>
> ***Table 3.** Performance comparison with full attention, using (Top) Llama-3.1-8B-Instruct and (Bottom) Mistral-Nemo-Instruct-2407. Full attention values are taken from the official leaderboards (BABILong performance for full-attention Nemo is not available).*
>
> |Llama-3.1|RULER 64k|RULER 128k|BABILong 64k|BABILong 128k|
> |-|:-:|:-:|:-:|:-:|
> |Full Attention|**84.7**|77.0|49.0|39.0|
> |**REFORM (Ours)**|84.5|**84.9**|**57.0**|**53.4**|
>
> |Mistral-Nemo|RULER 64k|RULER 128k|
> |-|:-:|:-:|
> |Full Attention|46.8|19.0|
> |**REFORM (Ours)**|**79.9**|**81.1**|
>
> ---
> **[W2-1] Implementation Details and Flash Attention Compatibility**
>
> **All efficiency experiments were done using Flash Attention 2** (see L539), and we will make this clearer in the captions of relevant tables. Since Flash Attention does not provide attention weights, we re-computed them using PyTorch matmul over the Q/K states. To minimize overhead, we limited these computations to Q vectors from the final 128 tokens of each chunk for H2O (see L230–L231), rather than using the full attention matrix.
>
> The prefill latency comparisons in Table 4 (and Table 6(b) in the main text) highlights the empirical effect of the implementation. Specifically, StreamingLLM does not require the additional score computation, while InfiniPot and REFORM do. While InfiniPot shows some increased latency, **REFORM shows reduced latency even with the additional operations, thanks to the computation savings from the early-exit strategy**. Furthermore, the latency breakdown measurements show that the **recomputation overhead for REFORM is minimal**.
>
> ***Table 4.** Prefill time (seconds) on Mistral-Nemo-Instruct-2407, single H200 GPU.*
>
> ||256k|512k|1M|
> |-|:-:|:-:|:-:|
> |StreamingLLM|30.59|68.22|143.57|
> |InfiniPot|35.77|73.67|149.64|
> |**REFORM (Ours)**|**26.24**|**53.68**|**108.64**|
> |-  Compress + Gather|25.84|53.28|108.24|
> |-  Recompute|0.40|0.40|0.40|
>
> ---
> **[W2-2] Complexity analysis**
>
> In Table 6(b) in the main text and Table 4 in the rebuttal, we present empirical analysis on the latency and memory requirements for each method. Here, we discuss the time- and memory-complexity of different approaches.
>
> Recurrent baselines such as StreamingLLM have a time complexity of O(L), where L is the input length, assuming a fixed (or bounded) chunk size and query length. Each recurrence step involves a constant amount of computation, and the total number of steps scales linearly with the input length. Their memory complexity is O(1), since only a fixed-size KV cache is maintained throughout.
>
> In contrast, random-access baselines like InfLLM incur a time complexity of O(L^2), primarily due to periodic cache lookups across the full input length. These methods also require O(L) memory, as they store the entire KV cache in memory. This memory burden is significant, often requiring CPU offloading, which further increases latency.
>
> REFORM maintains a time complexity of O(L) through its recurrent chunked processing. Although it has an O(L) memory cost due to storing token-level embeddings, these embeddings are very small in practice. Moreover, the early-exit strategy significantly reduces memory and compute requirements. Consequently, **REFORM achieves faster speed and lower memory usage than both baselines** (as demonstrated in Table 6(b) of the main text), highlighting the practical efficiency of our method.
>
> ---
> **[Q1] Clarifications of attention head selection.**
>
> We thank the reviewer for pointing this out and will revise the manuscript for better clarity. The head selection process is as follows:
>
> **Step 1: Data preparation.** We construct two synthetic tasks for attention head selection: a pattern matching task and a multi-hop question answering task. For the pattern matching task, we insert sentences into the WikiText corpus with the format: “The value corresponding to the id {key} is {value},” where both the key and value are random 10-character ASCII strings. For the multi-hop QA task, we use passages from HotPotQA that contain the information required to answer a given question. The tokens corresponding to this key information are labeled as “golden tokens.”
>
> **Step 2: Context encoding.** We apply recurrent chunked forwarding using H2O to each sample in the dataset, extracting token-level QKV embeddings from every layer and attention head.
>
> **Step 3: Significance score computation.** For each QKV head in each layer, we compute a token-wise significance score. This is done by (1) calculating cosine similarity between context tokens and question tokens, (2) aggregating the similarity scores over the question tokens via max-pooling, and (3) smoothing the scores using mean pooling across a 20-token window.
>
> **Step 4: Performance measurement and head selection.** We evaluate the effectiveness of each attention head using the Mean Normalized Rank (MNR) score, as described in Appendix C.1. Based on these scores, we select four heads: two that perform best on the pattern matching task, and two that perform best on the multi-hop QA task.
>
> We will revise the manuscript to incorporate this explanation more clearly.

---

> > ### Author Response · Authors · 2025-08-05
> > **A Gentle Reminder**
> >
> > Dear Reviewer gpyK,
> >
> > Thank you again for your time and effort in reviewing our paper.
> >
> > We kindly remind you that we only have a few days in the discussion period. We just wonder whether there is any further concern and hope to have a chance to respond before the discussion phase ends.
> >
> > Many thanks,
> >
> > Authors

---

> > > ### Comment · Reviewer_gpyK · 2025-08-07
> > > **Thank you for the rebuttal and additional experiments.**
> > >
> > > Thank you for the additional clarification re complexity analysis and the comparison against SnapKV.
> > >
> > > I will re-iterate that the claim that the approach is inference-only hides the non-trivial amount of computation needed to select the relevant attention heads. Therefore, the claim that it is apples-to-apples comparable to SnapKV is misleading, although i concede that it is less expensive than training.

---

> > > > ### Author Response · Authors · 2025-08-07
> > > > **Follow-up Response**
> > > >
> > > > Dear Reviewer gpyK,
> > > >
> > > > Thank you very much for the thoughtful follow-up. We sincerely appreciate your continued engagement and address your latest comment below.
> > > >
> > > > ---
> > > > **1. Revision on the positioning.**
> > > >
> > > > We acknowledge that the terms *"training-free"* or *"inference-only"* could be misleadingly interpreted as excluding any form of offline computation. To avoid ambiguity, we will revise the manuscript to clarify our positioning as follows:
> > > >
> > > > > "REFORM can be directly applied to existing models without modifying their parameters."
> > > >
> > > > This aligns with the framing adopted in works such as MInference (which uses a similar offline search strategy), which state:
> > > >
> > > > > "Our proposed technique can be directly applied to existing LLMs without any modifications to the pre-training setup or additional fine-tuning."
> > > >
> > > > We focused our comparisons on strong baselines that, like REFORM, can be applied without modifying the model's weights.
> > > >
> > > > While REFORM does involve computation for selecting a small number of relevant attention heads, we emphasize that our head selection process is conducted offline, requires no gradient computation, and is not specific to any downstream task.
> > > >
> > > > ---
> > > > **2. Comparison against approaches that utilize offline computation.**
> > > >
> > > > Furthermore, our experiments show that **REFORM substantially outperforms other methods that also employ head or pattern selection, such as Duo-Attention and MInference**, within the context window (see Table 2 in the rebuttal). This demonstrates that REFORM delivers stronger performance even compared to methods that involve similar types of offline computation.
> > > >
> > > > ---
> > > > **3. Remarks on the amount of offline computation.**
> > > >
> > > > We emphasize that the **computational overhead of head selection is minimal in practice**. The selection is performed only once using synthetic inputs of 8k tokens (see L514), and the chosen heads are reused across all inference runs, independent of the downstream task or input domain. By selecting heads using short inputs and reusing them across long-context inference, REFORM keeps the additional computation minimal.
> > > >
> > > > Quantitatively, the entire head selection procedure requires roughly the same amount of computation (in FLOPs) as processing just **two 1M-token inputs** with H2O. Given that this cost is incurred only once and the selected heads are used across all future usage, we believe it is a **practical and negligible overhead** compared to the benefits REFORM provides at inference time.
> > > >
> > > > ---
> > > > **4. Performance under apples-to-apples comparison.**
> > > >
> > > > Finally, we note that **REFORM often outperforms baselines even without head selection**. The results below (rearranged from Tables 2 and 6(a) in the main text) show that REFORM remains competitive even with randomly selected heads or poorly selected heads, showcasing that it remains effective even without the offline computation.
> > > >
> > > > ***Table 5.** Impact of head selection. (Mistral-Nemo-Instruct-2407).*
> > > >
> > > > || RULER 300k | BABILong 512k |
> > > > |-|:---:|:---:|
> > > > |REFORM (Selected Heads)| 84.6 | 47.6 |
> > > > |REFORM (Random Heads)| 80.3 | 43.0 |
> > > > |REFORM (Worst Heads)| 44.7 | 22.8 |
> > > > |Best Baseline| 24.9 | 13.6 |
> > > >
> > > > ---
> > > >
> > > > We hope this further clarifies the scope and positioning of REFORM, and we truly appreciate your constructive feedback throughout the review process.
> > > >
> > > > Please share your thoughts, and let us know especially if there are any additional concerns. We would be grateful for the opportunity to discuss further.
> > > >
> > > > Best regards,
> > > >
> > > > Authors

---

### Official Review · Reviewer_fog1 · 2025-07-03

**Clarity:** 3
**Significance:** 2
**Originality:** 2
**Rating:** 4
**Confidence:** 3

**Summary:**

The paper proposes REFORM, a two-phase approach for inferencing long-context LLMs. It first incrementally processes input chunks with a compressed kv cache with cross-layer context embedings and early exits strategy. It then gathers essential tokens and selectively recomputes KV cache. It improves results on several important benchmark such as RULER, inifinity-bench, RepoEval and MM-NIAH. For example, it achieves 52% and 34% performance gains on RULE and BABILong at 1M context length compared to baselines.

**Questions:**

(1) Could the author adds larger models result (e.g. Qwen-32B)?
(2) Could the author discuss more on token eviction? Currently the reviewer understands it is from H2O. How would this affect the paper result?
(3) Can the author adds the similarity search result and MNR scores (Table 1 and Figure 2) on Llama8B as well, and discuss how different models could affect the method?
(4) In line 331, the author mentions using 129-token windows, and using 5-tokens would significantly degrades performance. Could the author adds more numbers on how this change with respect to the window size?

**Ethical Concerns:**

["NO or VERY MINOR ethics concerns only"]

**Final Justification:**

As the original comment points out, the paper is generally in a good position. The author rebuttal is also reasonable, and thus I will keep my positive scores.

**Limitations:**

Yes.

**Paper Formatting Concerns:**

No major formatting issues.

**Quality:**

3

**Strengths And Weaknesses:**

Strength:
- The paper is well written and easy to follow: the motivation is clear, method is intuitive, and Figure 1 is illustrative of the proposed method.
- The problem is important, addressing the efficiency at extremely long (e.g. 1M) context length inference.
- The results are great, comprehensive (on multiple benchmarks). The ablation is throughout.

Weakness:
- Please consider adding larger models, e.g. 32B size.

---

> ### Author Rebuttal · Authors · 2025-07-30
>
> Dear reviewer fog1,
>
> We sincerely appreciate your efforts in reviewing our manuscript. We respond to each comment in the following content.
>
> ---
> **[W1, Q1] Results with Larger Models**
>
> As requested, we have included additional results using the Qwen2.5-32B-Instruct model on the RULER and BABILong benchmarks. As shown in Table 1, **REFORM consistently outperforms both H2O and InfiniPot** across all evaluation settings, demonstrating the scalability and robustness of our method on larger models.
>
> ***Table 1.** Performance of Qwen2.5-32B-Instruct.*
>
> ||RULER Single 300k|RULER Multikey 300k|RULER Multivalue 300k|RULER Multiquery 300k|BABILong 256k|
> |-|:-:|:-:|:-:|:-:|:-:|
> |H2O|38.7|2.7|14.0|6.0|31.0|
> |InfiniPot|69.3|19.3|40.0|74.0|54.2|
> |**REFORM (Ours)**|**100.0**|**90.0**|**96.0**|**100.0**|**67.6**|
>
>
> ---
> **[Q2] Analysis of Eviction Strategies**
>
> We appreciate the request for further analysis of eviction strategies. Table 2 compares REFORM’s performance using alternative eviction methods (StreamingLLM and TOVA), in addition to our primary method (H2O). While H2O yields the highest accuracy, alternative methods still significantly outperform baseline approaches. These results underscore the **flexibility of REFORM** with respect to the eviction strategy. (Refer also to Table 6(a) and Lines 318–322 of the manuscript.)
>
>
> ***Table 2.** Performance with different eviction methods (Mistral-Nemo-Instruct-2407)*
>
> ||RULER 300k|BABILong 512k|
> |-|:-:|:-:|
> |REFORM w/ H2O (Our choice)|**84.6**|**47.6**|
> |REFORM w/ StreamingLLM|82.7|44.6|
> |REFORM w/ TOVA|81.4|46.8|
> |Best Baseline|24.9|13.6|
>
>
>
> ---
> **[Q3] Similarity Search and MNR Results for Llama Models**
>
> In Table 3, we showcase the similarity search results for Llama-3.1-8B-Instruct. As with Mistral-Nemo (Table 1 and Table 7 in the manuscript), we observe that **cosine similarity using either Q, K, or V embeddings often yields the best performance**, outperforming alternatives such as attention scores or cosine similarity using the hidden states.
>
> Although we cannot share the MNR score distribution plots due to NeurIPS policy (image sharing restrictions), the observed **trends are consistent with our previously reported results**. Specifically, the top-performing heads in the pattern matching task tend to cluster at the early and the final layers, while top-performing heads for the multi-hop QA task tend to cluster in the middle layers.
>
> These findings indicate that **attention head significance patterns are consistent across architectures**.
>
>
> ***Table 3.** Similarity search results for (top) pattern matching task and (bottom) multi-hop QA task, measured with Llama-3.1-8B-Instruct.*
>
> |Embedding Type|Dim.|Top-1|Top-2|Top-3|Avg.|
> |-|:-:|:-:|:-:|:-:|:-:|
> |Attention|128|0.92|0.99|1.00|0.97|
> |Cosine-HS|4096|1.54|1.64|1.68|1.62|
> |Cosine-Q|128|1.21|1.24|1.46|1.30|
> |Cosine-K|128|1.23|1.30|1.43|1.32|
> |Cosine-V|128|**0.89**|**0.91**|**0.92**|**0.91**|
>
> |Embedding Type|Dim.|Top-1|Top-2|Top-3|Avg.|
> |-|:-:|:-:|:-:|:-:|:-:|
> |Attention|128|6.65|6.82|6.94|6.80|
> |Cosine-HS|4096|6.87|6.94|7.01|6.94|
> |Cosine-Q|128|6.17|**6.19**|**6.24**|**6.20**|
> |Cosine-K|128|6.81|7.30|7.58|7.23|
> |Cosine-V|128|**6.06**|6.50|6.57|6.38|
>
>
> ---
> **[Q4] Effect of Aggregation Window Size**
>
> In Table 4,  we conducted a more detailed ablation study on the aggregation window size. Performance improves with increasing window size up to 129, after which it begins to saturate or slightly degrade. This suggests that our original selection of 129 is a balanced and effective kernel size.
>
> ***Table 4.** Impact of Aggregation Window Size (Mistral-Nemo-Instruct-2407)*
>
> |Window Size|RULER (300k)|Babilong (512k)|
> |-|:-:|:-:|
> |5|18.4|36.8|
> |9|28.9|40.2|
> |17|39.4|45.8|
> |33|67.1|45.4|
> |65|70.7|47.4|
> |129|**84.6**|**47.6**|
> |257|84.3|44.4|

---

> > ### Comment · Reviewer_fog1 · 2025-08-01
> >
> > Thank you for the rebuttal! I will keep my score to recommend acceptance.

---

> > > ### Author Response · Authors · 2025-08-05
> > > **Thank You For The Acknowledgement**
> > >
> > > Dear Reviewer fog1,
> > >
> > > Thank you very much for acknowledging our rebuttal and recommending acceptance. We are sincerely grateful for your thoughtful and constructive review, which has significantly contributed to improving the clarity and rigor of our paper.
> > >
> > > Should you have any additional questions or comments during the discussion phase, we would be happy to engage further.
> > >
> > > Many thanks,
> > >
> > > Authors

---

### Official Review · Reviewer_NqoB · 2025-07-06

**Clarity:** 3
**Significance:** 3
**Originality:** 3
**Rating:** 4
**Confidence:** 4

**Summary:**

REFORM is a novel inference framework for efficiently handling extremely long contexts in large language models. It employs a two-phase approach: first, it processes input incrementally while compressing the KV cache, building cross-layer embeddings, and using early exits to enhance efficiency. Second, it selects essential tokens through similarity matching and selectively recomputes the KV cache. REFORM outperforms prior methods, achieving over 52% and 34% performance gains on RULER and BABILong at 1M context length. It also excels on Infini-Bench, RepoEval, and MM-NIAH benchmarks, while reducing inference time by 30% and peak memory usage by 5%, offering both speed and accuracy.

**Questions:**

See above.

**Ethical Concerns:**

["NO or VERY MINOR ethics concerns only"]

**Final Justification:**

Author's rebuttal addresses most of my concerns. I'll keep my original score leaning towards acceptance.

**Quality:**

3

**Strengths And Weaknesses:**

Strengths:
- The compression and reconstruction method is pretty interesting. It shares the similar spirit of [1] for fixing linear attention forgetting problem.
- The compress-gather-recompute pipeline is well-designed. It combines multiple technologies of H2O(compress), LayerSkip(early exit strategy), ShadowKV(cosine similarity), recomputation together to build a good sparse implementation. The combination is technically sound.
- It evaluates on a wide range of models and the ablation experiments is well designed for understand the contribution of different components.
- Performance improvement is significant and the implementation and setting is very practical.

Weaknesses:
- The evaluations are focused on retrieval tasks (including infinibench). It's understandable it will perform well since there's a query-based reconstruction phase. However, we suggest evaluation on more complicated tasks, such as GSM-Infinite[2], will make it more straightforward to see how the proposed method differs from RAG. (Now, this paper includes a comparison with RAG (8K budget), if only works well on retrieval tasks, considering that RAG does not even require a prefilling of LLM; therefore, it's not entirely convincing.)
- Lack of discussion and comparisons of several important work in dynamic KV access, e.g. Quest[3], MagicPIG[4], and prefilling acceleration: LM-infinite. Especially a comparison of iteratively applying SnapKV (https://arxiv.org/abs/2404.14469) [Iteratively prefill chunks of tokens into LLM, each time we use SnapKV to select a subset of previous tokens]. These comparisons will enhance and clarify the contribution of the work.
- The proposed method needs to reconstruct KV cache when a query comes. The reported re-computation budget is 8k-16k. Considering the settings in this paper, typically long-context and short output, this reconstruction overhead is significant. Thus, a more detailed efficiency analysis is necessary. For example, time for the first token, time per output token on different input lengths.
- All Efficiency Analysis are generating 10 tokens conditioned on 256k/1M inputs, therefore most of time is spending on prefilling phase. However, with LLM development with reasoning models' outstanding performance, decoding phase would generate more tokens than 10 tokens. The performance of REFORM on decoding phase is unknown and unclear. A detailed time table with sperate prefill/decoding would help a lot.
- The paper lacks theoretical analysis for why the selected attention heads perform well , and the compression-recomputation trade-offs. Can authors share the reasoning of selecting heads?


[1] Simran Arora, Aman Timalsina, Aaryan Singhal, Benjamin Spector, Sabri Eyuboglu, Xinyi Zhao, Ashish Rao, Atri Rudra, Christopher Ré. Just read twice: closing the recall gap for recurrent language models.

[2] Yang Zhou, Hongyi Liu, Zhuoming Chen, Yuandong Tian, Beidi Chen. GSM-Infinite: How Do Your LLMs Behave over Infinitely Increasing Context Length and Reasoning Complexity?

[3] Jiaming Tang, Yilong Zhao, Kan Zhu, Guangxuan Xiao, Baris Kasikci, Song Han. Quest: Query-Aware Sparsity for Efficient Long-Context LLM Inference.

[4] Zhuoming Chen, Ranajoy Sadhukhan, Zihao Ye, Yang Zhou, Jianyu Zhang, Niklas Nolte, Yuandong Tian, Matthijs Douze, Leon Bottou, Zhihao Jia, Beidi Chen. MagicPIG: LSH Sampling for Efficient LLM Generation.

---

> ### Author Rebuttal · Authors · 2025-07-30
>
> Dear reviewer NqoB,
>
> We sincerely appreciate your efforts in reviewing our manuscript. We respond to each comment in the following content.
>
> ---
> **[W1] Evaluations on GSM-Infinite**
>
> Following your suggestion, we evaluated REFORM and RAG variants on the 'medium' split of GSM-Infinite. As shown in Table 1, **REFORM consistently outperforms both sparse and dense RAG baselines**, demonstrating its ability to handle more complex reasoning tasks beyond simple retrieval.
>
> ***Table 1.** GSM-Infinite ('medium' split) evaluation with Llama-3.1-8B-Instruct. RAG methods use an 8k retrieval budget and REFORM uses an 8k recomputation budget. Responses were generated without chain-of-thought reasoning for efficiency. Details follow Appendix C.3.*
>
> ||ops_2|||ops_4|||ops_8|||
> |-|:-:|-|-|:-:|-|-|:-:|-|-|
> ||32k|64k|128k|32k|64k|128k|32k|64k|128k|
> |Sparse RAG|26.6|19.7|13.7|7.8|6.8|5.0|9.8|7.7|5.1|
> |Dense RAG|16.8|13.1|10.4|7.2|7.5|5.6|8.8|7.3|4.3|
> |**REFORM (Ours)**|**56.8**|**41.5**|**36.6**|**9.4**|**9.7**|**11.1**|**11.2**|**10.2**|**10.6**|
>
>
>
> ---
> **[W2] Comparison with Related Works**
>
> We compared REFORM to the suggested recurrent SnapKV baseline in Table 2. Notably, **REFORM significantly outperforms SnapKV across multiple tasks**, demonstrating its effectiveness.
>
> ***Table 2.** Comparison with recurrent SnapKV (Llama-3.1-8B-Instruct).*
>
> |  | RULER 300k | RULER 500k | RULER 1M | BABILong 256k | BABILong 512k | BABILong 1M | Infinite-Bench |
> |---|:---:|:---:|:---:|:---:|:---:|:---:|:---:|
> | SnapKV | 62.5 | 59.1 | 57.8 | 36.0 | 31.6 | 25.6 | 43.7 |
> | **REFORM (Ours)** | **82.0** | **82.0** | **79.0** | **46.0** | **46.6** | **43.8** | **53.2** |
>
> Before discussing other related works, we would like to emphasize that our main research goal is to propose an efficient, training-free **context window extension method** for large foundation models. Quest and MagicPIG **tackle a different problem** to improve efficiency **within the model's native context window** (e.g. <128k for Llama-3.1-8B-Instruct). We did not compare them in the paper as they cannot be used for inputs that are longer than the model's context limit.
>
> Furthermore, Table 3 in the MagicPIG paper reports that Quest and MagicPIG scores 74.9 and 81.7 respectively on RULER 96k using Llama-3.1-8B-Instruct. In contrast, REFORM achieves 84.9 at 128k and 82.6 at 200k with the same model, suggesting that it outperforms these methods, particularly in longer contexts.
>
> LM-Infinite employs a lambda-shaped mask to extend context, similar to StreamingLLM (a baseline in our paper). It optionally attends to prior tokens outside the lambda-mask for retrieval, but achieves only 70.3% accuracy on simple pass key retrieval tasks at 6k (Llama-2, 1.5x context limit), while REFORM reaches 100% at 1M (Llama-3.1, 8x context limit). Moreover, LM-Infinite’s full cache retention will lead to high memory requirements and offloading latency, similar to InfLLM.
>
> We believe that these comparisons and discussions further clarify REFORM’s positioning in the long-context inference space, and will add them to the revised manuscript.
>
> ---
> **[W3, W4] More Detailed Latency Analysis**
>
> Here, we extend our latency analysis to three context lengths (previously only 256k), and separately present the pre-fill and decoding latency in Table 3 and Table 4.
>
> In Table 3, InfiniPot and InfLLM shows slower processing speed compared to StreamingLLM due to the additional computation associated with dynamic cache eviction and retrieval. On the other hand, **REFORM consistently shows faster prefill time compared to all baselines, thanks to the early-exit optimization**. Further latency breakdown of the Compress+Gather stage and Recompute stage suggests that the **recomputation overhead is minimal**, taking under 1.52% of the total prefill time.
>
> Table 4 shows a similar trend for the decoding time. InfiniPot and InfLLM shows slower decoding time compared to StreamingLLM due to the additional operations. On the other hand, **REFORM also shows improved decoding speed** as the generation is conditioned on KV cache created with a small recomputation budget (8k), unlike the baselines that condition on the full compressed cache (32k). (Note that these budgets are the hyperparameters used for the Mistral-Nemo experiments in our main text.)
>
> ***Table 3.** Time to first token (seconds) measurements (Mistral-Nemo-Instruct-2407, single H200, averaged over 20 runs).*
>
> ||256k|512k|1M|
> |-|:-:|:-:|:-:|
> |StreamingLLM|30.59|68.22|143.57|
> |InfiniPot|35.77|73.67|149.64|
> |InfLLM|95.71|213.23|474.96|
> |**REFORM (Ours)**|**26.24**|**53.68**|**108.64**|
> |-  Compress + Gather|25.84|53.28|108.24|
> |-  Recompute|0.40|0.40|0.40|
>
> ***Table 4.** Time per output token (seconds) measurements (Mistral-Nemo-Instruct-2407, single H200, averaged over 20 runs and 200 tokens generated per measurement).*
>
>
> ||256k|512k|1M|
> |-|:-:|:-:|:-:|
> |StreamingLLM|0.111|0.111|0.111|
> |InfiniPot|0.256|0.256|0.256|
> |InfLLM|0.259|0.267|0.329|
> |**REFORM (Ours)**|**0.040**|**0.040**|**0.040**|
>
> ---
> **[W5-1] Discussion on Compression-Recomputation Trade-Off**
>
> While cache recomputation overcomes the forgetting problem, it introduces an additional recomputation overhead, leading to a trade-off. Latency breakdown in Table 3 suggests that **the recomputation overhead is marginal**, while our extensive evaluations in the main text demonstrate that the **recomputation gives significant improvements in downstream performance**.
>
> ---
> **[W5-2] Theoretical Analysis on Head Selection**
>
> Our attention head selection is **motivated by findings from mechanistic interpretability literature**, which show that different heads specialize in different functions [1,2]. Some heads are more useful than others for tasks like pattern recognition or reasoning. We leverage this diversity by selecting only a few high-performing heads, both for better identification performance and to keep the size of token-level embeddings tractable (including all heads would be memory-intensive). Our method thus aims for functional specialization as well as computational efficiency.
>
> Although we use cosine similarity rather than attention weights to evaluate heads, **our approach shares key insights with existing interpretability work**. Prior studies show that lower-layer heads often detect syntactic or structural features, mid-layer heads handle semantic reasoning, and upper-layer heads guide output generation [3]. This aligns with our empirical findings: the most useful heads for pattern matching tend to appear in lower and upper layers, while those for multi-hop QA (i.e. semantic understanding) concentrate in the mid-layers. These trends are reflected in our MNR visualizations (Fig. 2 and Fig. 4). Also, Skean et. al. [4] suggests that the middle layer representations are more useful than the final layer representations for 32 MTEB (Massive Text Embedding Benchmark) tasks, similar to what we observe for the multi-hop QA task.
>
> Furthermore, **our choice of the head identification tasks is rooted in information retrieval literature**, which shows that hybrid retrievers [5,6] combining sparse (structural) and dense (semantic) retrievers outperforms either alone. Inspired by this, we use two distinct head evaluation tasks: pattern matching and multi-hop QA. The former identifies heads that capture structural similarity, while the latter targets heads that encode semantic relevance. This dual-task strategy helps us select a complementary set of heads that are robust across diverse long-context scenarios.
>
> Together, these design choices provide high token identification accuracy while maintaining efficiency.
>
> [1] Voita et. al., Analyzing multi-head self-attention: Specialized heads do the heavy lifting, the rest can be pruned
>
> [2] Ren et. al., Identifying semantic induction heads to understand in-context learning
>
> [3] Zheng et. al., Attention Heads of Large Language Models: A Survey
>
> [4] Skean et. al., Layer by Layer: Uncovering Hidden Representations in Language Models
>
> [5] Bruch el. al., An analysis of fusion functions for hybrid retrieval.
>
> [6] Chen el. al., Out-of-domain semantics to the rescue! zero-shot hybrid retrieval models.

---

> ### Author Response · Authors · 2025-08-05
> **Thank You For The Acknowledgement**
>
> Dear Reviewer NqoB,
>
> Thank you very much for acknowledging our rebuttal. We are grateful for the time and effort you put in the review and reading through the rebuttals, which has greatly contributed to improving the clarity and soundness of our paper.
>
> Through the rebuttal, we believe that we have clarified your concerns successfully. We would greatly appreciate any further comments or insights you may have after reading our response, especially if any of your earlier concerns remain. We would be glad to engage in further discussion during the remainder of the rebuttal period.
>
> Many thanks,
>
> Authors

---

> > ### Comment · Reviewer_NqoB · 2025-08-05
> > **Concerns resolved**
> >
> > Thanks authors for the detailed response and added experiments on new benchmark like gsm-inf. I'll keep my support to the paper.

---

> > > ### Author Response · Authors · 2025-08-09
> > > **Thank you for the comments**
> > >
> > > Dear Reviewer NqoB,
> > >
> > > Thank you again for providing constructive feedback and recommending acceptance. We are sincerely grateful for your thoughtful and constructive review, which has significantly contributed to improving our paper.
> > >
> > > Best regards,
> > >
> > > Authors

---

### Decision · Program_Chairs · 2025-09-17

**Decision:**

Accept (poster)

**Comment:**

This paper introduces REFORM, a novel inference framework to handle long context through two phases: 1) incrementally processes input chunks while maintaining a compressed KV cache, and 2) identifies and gathers essential tokens via similarity matching and recomputes the KV cache selectively. REFORM outperforms baselines on Infinity-Bench, RepoEval, and MM0NIAH, and achieves performance gains of over 52% and 34% on RULE and BABILong at a 1M context length. REFORM also reduces the inference time by 30% and peak memory usage by 5%.

Strengths:
+ A well-designed compress-gather-recompute pipeline that combines multiple technologies, including H2O (compress), LayerSkip (early exit strategy), and ShadowKV (cosine similarity).
+ REFORM addresses the efficiency at extremely long context (1M) with excellent results on multiple benchmarks and thorough ablation.
+ Thorough and strong rebuttal that addresses the questions and concerns from all reviewers.

Weaknesses:
- No major weaknesses, only two limitations explained by the authors during the rebuttal: 1) the effectiveness of REFORM depends on the gather stage's ability to identify informative tokens that could affect edge cases, and 2) redundant attention score computation in the current implementation.

Overall, REFORM is a sound and novel framework that outperforms other relevant baselines. The authors provided a comprehensive rebuttal that addressed all concerns raised by the reviewers. I reviewed the rcHLb's concerns, and the authors offered clear and firm answers to highlight the key differences between REFORM and H2O, particularly that REFORM addresses the context forgetting issues of H2O. So, I would strongly suggest that this paper be accepted.